# Combined Model-Based Prediction for Non-Invasive Prenatal Screening

**DOI:** 10.3390/ijms232314990

**Published:** 2022-11-30

**Authors:** So-Yun Yang, Kyung Min Kang, Sook-Young Kim, Seo Young Lim, Hee Yeon Jang, Kirim Hong, Dong Hyun Cha, Sung Han Shim, Je-Gun Joung

**Affiliations:** 1Department of Biomedical Science, College of Life Science, CHA University, Seongnam 13488, Republic of Korea; 2Center for Genome Diagnostics, CHA Biotech Inc., Seoul 06135, Republic of Korea; 3CHA Future Medicine Research Institute, CHA Bundang Medical Center, Seongnam 13488, Republic of Korea; 4Department of Biomedical Engineering, Hankuk University of Foreign Studies, Yongin 17035, Republic of Korea; 5Department of Obstetrics and Gynecology, CHA Gangnam Medical Center, CHA University, Seoul 06135, Republic of Korea; 6Institute for Biomedical Informatics, CHA University School of Medicine, CHA University, Seongnam 13488, Republic of Korea

**Keywords:** non-invasive prenatal testing, next-generation sequencing, trisomy, copy number variations, clinical significance

## Abstract

The risk of chromosomal abnormalities in the child increases with increasing maternal age. Although non-invasive prenatal testing (NIPT) is a safe and effective prenatal screening method, the accuracy of the test results needs to be improved owing to various testing conditions. We attempted to achieve a more accurate and robust prediction of chromosomal abnormalities by combining multiple methods. Here, three different methods, namely standard Z-score, normalized chromosome value, and within-sample reference bin, were used for 1698 reference and 109 test samples of whole-genome sequencing. The logistic regression model combining the three methods achieved a higher accuracy than any single method. In conclusion, the proposed method offers a promising approach for increasing the reliability of NIPT.

## 1. Introduction

Most Organisation for Economic Co-operation and Development countries are struggling to develop countermeasures to social problems that lead to low fertility, aging, and population decline. In particular, the average age of childbirth among Korean women is over 30 years, and this age is increasing year by year, while the fertility rate has been continuously decreasing over the past five years [1]. Elderly pregnant women with high-risk ultrasound and maternal serum screening test results, a history of pregnancy with a chromosomally abnormal fetus, or a family history of disease are at risk for various complications, including congenital anomalies, gestational hypertension, placental premature abruption, and placenta previa; therefore, various first-trimester screenings are recommended [2,3].

In the past, ultrasound and maternal serum markers were used in the early pregnancy stages [4], and when additional screening tests were needed, invasive prenatal diagnostic tests, such as chorionic villus sampling (CVS), were performed. Non-invasive methods are needed to avoid the risk of miscarriage; hence, next-generation sequencing (NGS)-based non-invasive prenatal testing (NIPT) has been actively studied [5]. Two representative genomic methods for measuring chromosomal aneuploidy are quantitative read count-based methods and single-nucleotide polymorphism genotyping methods. Applying quantitative read counting-based methods, Down syndrome, Patau syndrome, Edward syndrome, and sex chromosome abnormalities can be diagnosed using the mass-parallel whole genome sequencing method [2,5].

NIPT is a non-invasive prenatal chromosomal abnormality test that safely and accurately detects chromosomal abnormalities in the fetus by analyzing fetal DNA present in the blood of pregnant women after 10 weeks of gestation using NGS [6]. In previous studies of NIPT, based on the positive predictive value, sensitivity, and specificity, the performance of trisomy 13 and 18 showed poor accuracy than that of trisomy 21 [7,8]. As it uses the pregnant woman’s blood, it is greatly affected by the pregnant woman’s conditions, such as age, body mass index (BMI), gestational week, number of fetuses, and confined placental mosaicism [5]. In particular, it is directly affected by factors, such as fetal fraction, which is important when analyzing NIPT results. The fetal fraction is the ratio of cell-free fetal DNA (cffDNA) in the maternal plasma to all cell-free DNA (cfDNA) [6]. It is estimated to be about 10% of free DNA in the mother’s plasma during 10 to 22 weeks [9] of gestation and is known to vary considerably between individuals. Due to the short size [10] and half-life [11,12] of cffDNA, various sample acquisition processes, such as sample collection, storage, and transport, are quite important. Additional data filtering is required when the sample undergoes degradation and hemolysis of red blood cells. Meanwhile, as NIPT is performed in early pregnancy, an improved experimental method is also required to minimize the occurrence of false positives and false negatives.

Many analytical methods depend on the statistical comparison between a sample of interest and a reference set of euploid control samples [13,14,15]. Attempts have also been made to construct an R-based analysis workflow that includes GC corrections, standard Z-scores, and normalized chromosome value (NCV) calculations [16,17]. WISECONDOR, which uses within-sample normalization techniques, and NIPTeR, an open-source R package that enables fast NIPT analysis and simple but flexible workflow creation [18,19] including variation reduction, trisomy prediction algorithms, and quality control, are two examples. These tools allow users to account for variability in the NIPT data, calculate the control group statistics, and predict the presence of trisomies. In this study, we maximized the predictive performance by combining the results of multiple tools. In addition, we investigated the relationship between several parameters and the clinical characteristics of the subject (pregnant woman).

## 2. Results

### 2.1. Basic Information on the Study Samples

The samples in our study were collected at 12 weeks of pregnancy (±1.32, gestational age) on an average and the mean BMI was 22.02 $\mathrm{kg}/m^{2}$(±3.42). The average maternal age was 36.63 (±3.51) years and 76.1% of the tested pregnant women belong to the advanced maternal age group, whose ages are greater than 35 years. This high-risk population is generally associated with poor obstetrical and perinatal outcomes [20]. A total of 1698 normal samples were used as controls for measuring the Z-scores, and the other 109 samples were used as subjects for our NIPT. These normal samples were selected based on the following criteria: confirmed by an independent tool as showing no chromosomal abnormalities and confirmed as normal on secondary tests such as CVS and amniocentesis. Among the test samples, 5 and 33 samples were identified to have trisomy 18 and 21, respectively, through the secondary test. The classification model was evaluated using test samples. The demographic information of the samples is summarized in Table 1.

### 2.2. Features of a Pregnanncy-Affecting Fetal Fraction

First, the effects of gestational age and BMI on fetal fraction were investigated using reference samples. The average fetal fraction of the samples collected during the first trimester (0–13 weeks) was 11.8% (median: 11.5) and that of samples collected in the second trimester (14–27 weeks) was 12.5% (median: 12.3) (Figure 1A). As in previous studies [21], fetal fraction increased as the age of pregnancy (weeks) increased (*p* = 0.044). The mean of fetal fractions was also calculated for the four groups and classified according to BMI: 13.2% (±3.62) in the underweight group (BMI < 18.5), 12.1% (±3.39) in the healthy group (18.5–25), 9.99% (±3.10) in the overweight group (25–30), and 8.91% (±2.67) in the obese group (≥30) (Figure 1B). Contrary to the gestational age, fetal fraction tended to decrease as BMI increased (*p* < 0.05 for all pairwise comparisons); therefore, our study confirmed that BMI and period of pregnancy can affect the fetal fraction [22].

### 2.3. Factors Affecting the Z-Score of NIPT

We examined whether the GC content, fetal fraction, and maternal age influenced the Z-score derived by each of the three methods. No relationships between GC content or fetal fraction and Z-score were not found because most normal samples had Z-scores near zero, but those in the samples identified with trisomies by a secondary test were correlated with Z-score (Figure 2A,B). The Z-scores obtained by STD (*r* = –0.33, *p* = 0.04) and within-sample reference bin (WSRB; *r* = –0.35, *p* = 0.03) were inversely correlated with the GC content. Conversely, the fetal fraction was directly proportional to the Z-scores for the three methods (STD, *r* = 0.63, *p* = 2.5 × 10^−5^; NCV, *r* = 0.63, *p* = 2.6 × 10^−5^; WSRB, *r* = 0.61, *p* = 4.8 × 10^−5^). Based on the maternal age, the average Z-scores of STD, NCV, and WSRB for the advanced maternal group were 1.52, 1.71, and 1.48, respectively, while those of the other groups were 0.42, 0.49, and 0.66, respectively (Figure 2C). Similar to the fetal fraction, the Z-score was proportional to the maternal age (STD, *p* = 0.031; NCV, *p* = 0.02; WSRB, *p* = 0.036). Therefore, GC content, fetal fraction, and maternal age are regarded as factors that can affect the Z-scores. Moreover, the BMI of pregnant women and gestational age associated with fetal fraction may indirectly affect them.

### 2.4. NIPT Results of the Three Methods

When the Z-score obtained by each method was 3.0, the sample was primarily considered to have a trisomy. Trisomy 13, 18, and 21 were identified (5, 9, and 33 by the STD; 5, 10, and 34 by NCV; 3, 9, and 33 by WSRB) (Figure 3A; Appendix A; Appendix A). The prediction accuracy of each method is summarized in Table 2. All three methods produced incorrect predictions of S004, S005, and S093 as trisomy 13, S007, S067, S077, and S078 as trisomy 18, and S091 and S092 as trisomy 21 (false-positive). In addition, normal samples, S107 and S108, were referred to as trisomy 13 by STD and NCV, and S028 was labeled as trisomy 18 by NCV. S029, with trisomy 21 identified by amniocentesis, was misclassified as a normal sample (false-negative) in all three methods, and S030, with trisomy 21, was also classified as a normal sample by the STD and WSRB methods (Figure 3B). NCV method had the highest number of false-positives but the lowest number of false-negatives, and the WSRB method resulted in the fewest false-positives; however, nine samples were identified as false-positives by all three methods.

### 2.5. Logistic Regression with Parameters Using Z-Scores, GC Content, and Fetal Fraction

To reduce the number of false-positives and increase the accuracy of prediction, we constructed a logistic regression model using three Z-scores as the basic parameters and GC content and fetal fraction as additional parameters. Logistic regression with basic parameters showed that one false-positive (S092) and two false-negatives (S029 and S030) occurred on chromosome 21. The logistic regression method was more accurate than each individual NIPT method (Table 3). When the GC content and fetal fraction, associated with the Z-score, were used as additional parameters, not only the false-positives and -negatives from logistic regression with basic parameters, but also three additional false-positives (S005, S092, and S109) were observed (Figure 4A). The area under the receiver operating characteristic curve of the logistic regression using three and five parameters was 0.99 and 0.98, respectively (Figure 4B); therefore, using only three Z-scores as the inputs is advantageous to obtain more precise results (Figure 4C).

## 3. Discussion

We aimed to reduce the number of false-positives, which requires re-testing that involves considerable effort and cost to confirm the reliability of the original results. In our study, we combined three methods based on Z-scores calculated by the difference in the read counts between the control group and test sample [13] into the logistic regression model, resulting in a more accurate prediction of aneuploidies, such as trisomies 13, 18, and 21. Each method determines whether a sample is aneuploid based on the Z-score acquired from individual algorithms, each having its own advantages and disadvantages. For example, the NCV method predicted the highest number of false-positives and the lowest number of false-negatives, but the WSRB method predicted the lowest number of false-positives in our samples among the three methods. Integrating the results of the three approaches improves the accuracy by adjusting the cutoff to detect aneuploidy and covering the advantages and disadvantages of each method.

The counts of predicted false-positives and -negatives of STD, NCV, and WSRB were 11 and 2, 12 and 1, and 9 and 2, respectively; however, the logistic regression resulted in an error of one false-positive and two false-negatives when performing cross-validation. Adding the fetal fraction and GC content as parameters generated three additional false-positives, and the incorporation of additional factors had no benefit in reducing the false-positives.

Our proposed method is limited in its ability to perfectly classify the model. S092_chr21, a false-positive detected by our logistic regression algorithm, was confirmed to be euploid by karyotyping for amniocentesis. This sample had chromosome 21 Z-scores of greater than three for all three methods (STD, 4.59; NCV, 4.55; and WSRB, 4.10). This may be due to problems with the sample, such as confined placental mosaicism, maternal copy number variants, apoptotic tumor cells from a mother, DNA of a vanished co-twin, maternal aneuploidies, or structural abnormalities [23,24]. S029_chr21 and S030_chr21, obtained from identical pregnant women at the same time but differing in sequencing date, had the mosaicism of trisomy 21 identified by karyotyping of amniotic fluid cells. This suggests that the fetus may be a mosaic trisomy 21 because the cells in the amniotic fluid mostly originate from the fetus [25]. Missing a sample with this mosaicism is considered to be a fundamental limitation of the current cfDNA technology because the fetal cfDNA in maternal circulation originates mainly from cytotrophoblast [26]. In general, these samples are verified by performing a secondary test because the Z-scores are located near the boundary. Therefore, we failed to detect samples whose sequencing data showed a slight difference in the proportion of chromosome 21 to the reference set.

Our model improves the accuracy of detection and the number of pregnant women undergoing invasive procedures associated with high risk [27] can be decreased if this model is used for diagnosis. In subsequent studies, we aim to investigate the distinct features of a normal sample with a high Z-score, apply them to a model, and precisely identify a case with mosaicism, while ensuring a small number of false-positives.

## 4. Materials and Methods

### 4.1. Sample Preparation and Sequencing

Samples were collected between 9 and 23 weeks of gestation from patients attending the CHA Medical Center between January, 2019 and December, 2021. Peripheral blood samples (approximately 10 mL) were collected in cfDNA BCT tubes (Streck, Omaha, NE, USA) and whole blood samples were centrifuged at 1200× *g* for 10 min at 4 °C. Plasma was isolated from the maternal cells, transferred to microcentrifuge tubes, and centrifuged at 16,000× *g* for 10 min. The supernatant, including cfDNA, was separated from the residual cells, transferred to a new tube, and stored at −20 °C. CfDNA was extracted using the QIAamp Circulating Nucleic Acid Kit (Qiagen, Hilden, Germany), according to the manufacturer’s instructions. cfDNA was constructed using the Ion Plus Fragment Library kit (Thermo Fisher, Waltham, CA, USA), according to the manufacturer’s instructions. A total of 12 samples were loaded on an Ion 540 Chip Kit (version 2.0; Life Technologies, Carlsbad, CA, USA) at one time, yielding an average of 0.3 × sequencing coverage per nucleotide. The number of raw reads for each sample was >2 million.

### 4.2. Shallow NGS Data Analysis

Sequencing reads obtained from the Ion Torrent Suite software were aligned to the human reference genome (hg19) using the torrent mapping alignment program with default settings. The reads were filtered by mapping to a quality of 30 or higher and sorted by mapping positions using SAMtools (ver 1.10) [28]. The fetal fraction (%) of a sample was predicted using filtered read counts by the weighted rank selection criterion (WRSC) model of the seqFF tool [29]. We considered the testing result of a sample with a fetal fraction of 4% or more reliable.

### 4.3. NIPT Analysis

NIPT analysis was performed using NIPTeR (ver 1.0.2) [19] and WisecondorX (ver 1.2.4) [18]. In NIPTeR, the read counts were corrected based on GC contents for each of the 50 kb bins, and χ^2^ (chi-squared) based variation reduction was performed for the bins, which have a higher variability. The standard Z-score (STD) of the sample was calculated as the difference in read counts between the reference group and the analyzed sample using the average and standard deviation of those of the reference set for chromosomes 13, 18, and 21. NCV was computed by setting the maximum number of chromosomes used as the denominator to 9. In WisecondorX, the Z-score was determined through WSRB by setting the bin size to 50 kb. The sample with a Z-score of 3.0 or higher was considered to have trisomy for all methods.

### 4.4. Logistic Regression

A logistic regression model was employed to determine the decision boundary for classification by combining the methods. The Z-scores derived by the three methods as basic parameters, with GC content (%) and fetal fraction as additional parameters, were used as the input data. To evaluate the prediction accuracy, leave-one-out cross-validation was conducted for 109 samples. The sample with a probability of more than 0.5 was estimated to have trisomy, and all statistical analysis and graphing were implemented in R, ver 4.0.5.

## Figures and Tables

**Figure 1 ijms-23-14990-f001:**
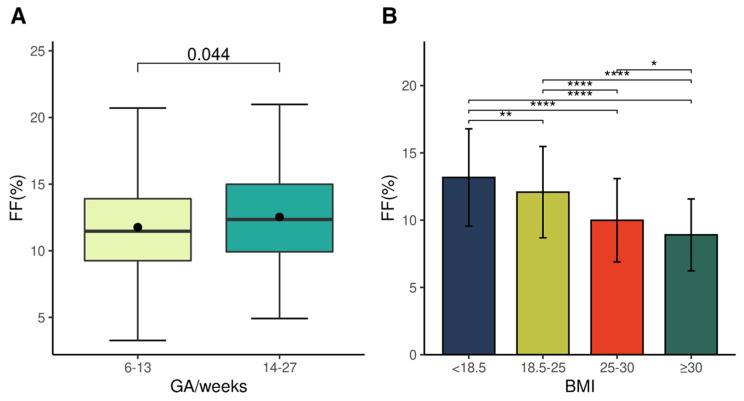
Correlations of fetal fraction (FF) with gestational age (GA) and body mass index (BMI) in the reference set (*n* = 1698). (**A**) Boxplots of FF (%) by GA (weeks). The middle point represents the average of FF. *p*-value is given at the top of the figure. (**B**) Mean of FF (%) by BMI. According to the BMI value, samples are divided into four groups: underweight (<18.5), healthy (18.5–25), overweight (25–30), and obese (≥30). * *p* < 0.05, ** *p* < 0.01, and **** *p* < 0.0001. FFs were calculated using the weighted rank selection criterion (WRSC) model in SeqFF.

**Figure 2 ijms-23-14990-f002:**
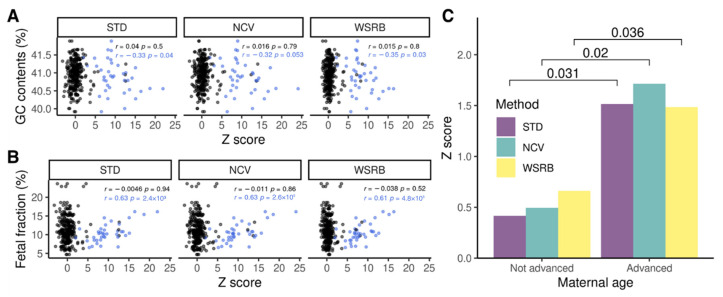
Correlations of Z-scores obtained using three methods (STD, normalized chromosome value [NCV], and WSRB) with GC content (GC, %), FF (%), and maternal age in 109 non-invasive prenatal testing (NIPT) samples. (**A**) Relationship between GC content (%) and Z-score. (**B**) Relationship between FF (%) and Z-score. A dot point represents a Z-score of a specific chromosome (chr 13, 18 or 21) from a sample. Therefore, each graph has 327 points displayed. Black represents a normal sample, and blue represents a trisomy sample identified by chorionic villus sampling (CVS) or amniocentesis. In each graph, *p*-value and r-value were calculated for normal and trisomy group, respectively. (**C**) Mean of Z-scores of three methods (STD, NCV, and WSRB) by maternal age. Advanced maternal age is equal to and older than 35.

**Figure 3 ijms-23-14990-f003:**
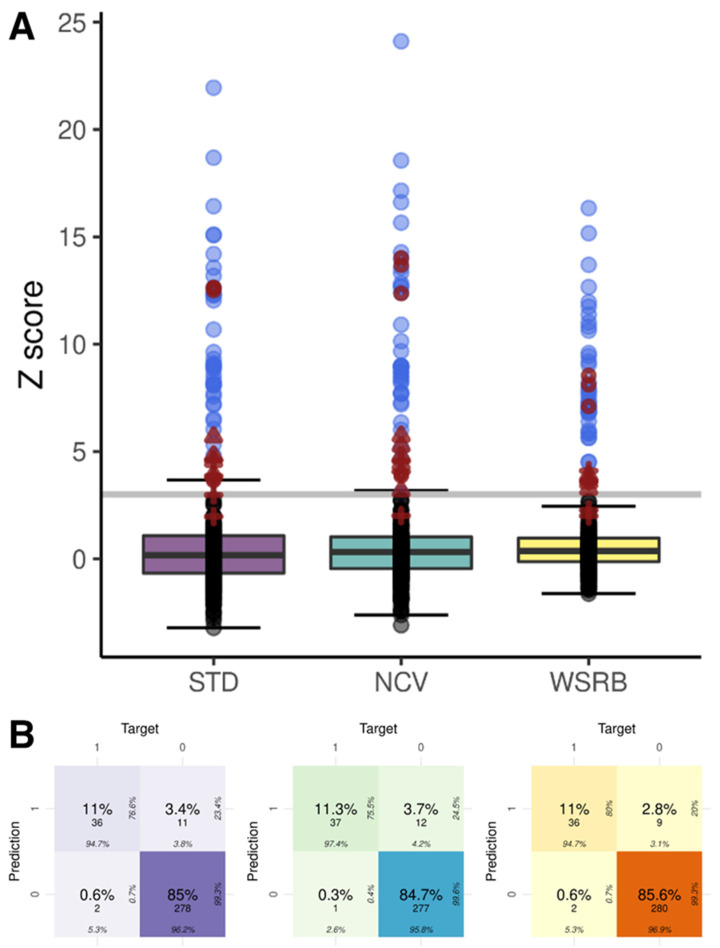
NIPT results of 109 samples obtained using three methods (STD, NCV, and WSRB). (**A**) Boxplots of Z-scores of three methods. Gray line represents the cutoff of detection of trisomy (Z-score = 3.0). An open red circle represents the chromosome 13 of a sample that differs between a NIPT result and a secondary test, an open red triangle represents chromosome 18 showing the difference, and a red plus represents chromosome 21 showing the difference. (**B**) Confusion matrix of prediction with Z-scores for three chromosomes (chr 13, 18, and 21) of three different methods. Value of 0 represents a normal (not trisomy) sample and 1 represents a trisomy. Graph colors are different according to the specific method (STD: purple, NCV: turquoise, WSRB: yellow).

**Figure 4 ijms-23-14990-f004:**
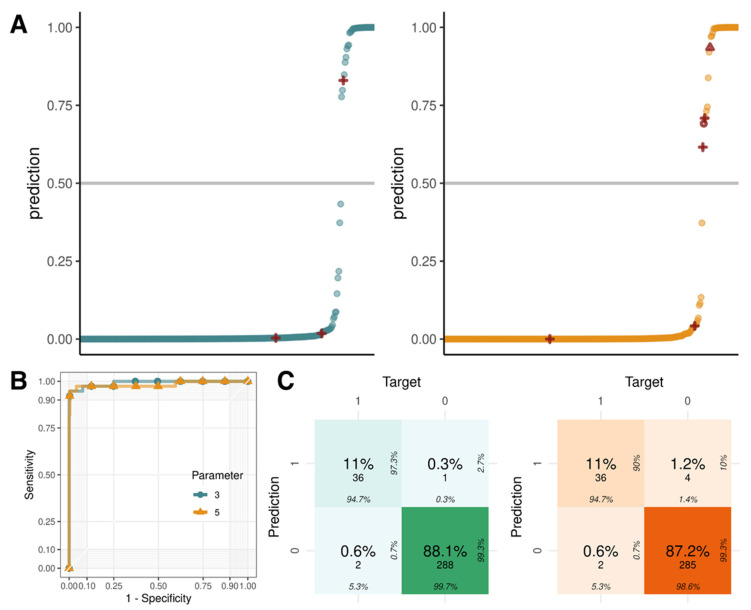
Logistic regression with the Z-scores, GC content (%), and FF (%) parameters of the three methods. (**A**) Logistic regression with three (STD, NCV, and WSRB) and five (STD, NCV, WSRB, GC, and FF) parameters. A point indicates the probability that a specific chromosome of a sample is a trisomy. A red mark represents incorrect prediction. (**B**) Receiver operating characteristic (ROC) curves of logistic regression with three and five parameters. (**C**) Confusion matrix of logistic regression with three and five parameters. Graph colors are different according to the parameter number (three parameters: green, five parameters: orange).

**Table 1 ijms-23-14990-t001:** Demographic characteristics of the reference and test samples.

Characteristic	Reference Set (*n* = 1698)	Test Set (*n* = 109)
GA at NIPT (weeks)		
First trimester (6–13)	1594	89
Second trimester (14–27)	104	20
Maternal age (years)		
20–29	54	3
30–39	1314	79
≥40	330	27
Advanced maternal age (≥35 years)	1291	85
BMI		
<18.5	140	12
18.5–25	1293	81
25–30	213	9
≥30	52	7
Pregnancy		
Singleton	1623	105
Twin	75	4
Secondary outcomes		
Trisomy 18		5
Trisomy 21		33

GA: gestational age.

**Table 2 ijms-23-14990-t002:** Statistics of the accuracy, sensitivity, specificity, positive predictive value (PPV), and negative predictive value (NPV) of the three methods (STD, NCV, and WSRB).

	Accuracy	Sensitivity	Specificity	PPV	NPV
STD	0.960	0.947	0.962	0.766	0.993
NCV	0.960	0.974	0.958	0.755	0.996
WSRB	0.966	0.947	0.969	0.800	0.993

**Table 3 ijms-23-14990-t003:** Statistics of the accuracy, sensitivity, specificity, PPV, and NPV of logistic regression with three and five parameters.

	Accuracy	Sensitivity	Specificity	PPV	NPV
3 parameters	0.991	0.947	0.997	0.973	0.993
5 parameters	0.982	0.947	0.986	0.900	0.993

## Data Availability

The data that support the finding of this study are available from the corresponding author, upon reasonable request.

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
