# Peer review of "Combined Model-Based Prediction for Non-Invasive Prenatal Screening"

_ijms, 2022, doi:10.3390/ijms232314990_

Round 1
Reviewer 1 Report
In the paper the author attempted to achieve a more accurate and robust prediction of chromosomal abnormalities by combining multiple methods. For this aim they used three different methods, namely standard Z-score, normalized chromosome value, and within-sample reference bin, for 1,698 reference and 109 test samples of whole-genome sequencing. They used the logistic regression model combining the three methods achieved a higher accuracy than any single method.
Major revision
- In this study, the mean of the gestational age is greater than 35 years old, so the pregnant women tested described a high-risk population. It means that in this study a seledted population is considered. I suggest to add this in the text ?
- How the population of the 1698 normal samples have been selected?
- IntroductioN lane 51: add some sentences about the test performance of the NIPT.
-In the text the authors suggested that the combined model described can be used for trisomies 13, 18 and 21. Moreover in the study, the test samples were only trisomies 18 and 21.
- The model proposed improves the accuracy of detection and the number of pregnant women undergoing invasive procedures associated with high risk results. But also the false negative is a problem. Can the authors comment about this?
Author Response
Reviewr #1
Thank you for the thoughtful comments and useful suggestions on our manuscript. We have revised the manuscript accordingly.
Comments and Suggestions for Authors
In the paper the author attempted to achieve a more accurate and robust prediction of chromosomal abnormalities by combining multiple methods. For this aim they used three different methods, namely standard Z-score, normalized chromosome value, and within-sample reference bin, for 1,698 reference and 109 test samples of whole-genome sequencing. They used the logistic regression model combining the three methods achieved a higher accuracy than any single method.
Major revision
- In this study, the mean of the gestational age is greater than 35 years old, so the pregnant women tested described a high-risk population. It means that in this study a seledted population is considered. I suggest to add this in the text?
Response: Thank you for the helpful suggestion. We added the details to Results section (in revised manuscript line 85): “The average maternal age was 36.63 (±3.51) years and 76.1% of the tested pregnant women belong to the advanced maternal age group, whose ages are greater than 35 years. This high-risk population is generally associated with poor obstetrical and perinatal outcomes. (https://doi.org/10.1186/s12884-020-2740-6)”.
- How the population of the 1698 normal samples have been selected?
Response: Thanks for a helpful comment. The population of the 1,698 normal samples have been selected based on the following criteria: confirmed by an independent tool as showing no chromosomal abnormalities and confirmed as normal on the secondary test. We added this to Results section (in revised manuscript line 90).
- IntroductioN lane 51: add some sentences about the test performance of the NIPT.
Response: As the referee commended, we added the sentences to Introduction section (in revised manuscript line 54): “In previous studies of NIPT (https://doi.org/10.1002/pd.5908, https://doi.org/10.1186/s12884-021-03570-6), based on the positive predictive value, sensitivity, and specificity, the performance of trisomy 13 and 18 showed poor accuracy than that of trisomy 21.”
-In the text the authors suggested that the combined model described can be used for trisomies 13, 18 and 21. Moreover in the study, the test samples were only trisomies 18 and 21.
Response: Thank you for your insightful comment. At the time of data correction, sample identified as trisomy 13 in both NIPT and secondary test could not be obtained because of its rareness. Our model tested samples for chromosome 13, 18, and 21, and on chromosome 13, all samples were accurately predicted as true negative.
- The model proposed improves the accuracy of detection and the number of pregnant women undergoing invasive procedures associated with high risk results. But also the false negative is a problem. Can the authors comment about this?
Response: Thank you for the detailed comment. As described in Discussion (in original manuscript line 205), S029 and S030 are false negative samples that have a mosaicism identified by karyotyping of amniotic fluid cells. We added the sentences about the limitation of detecting a mosaicism by NIPT to Discussion section (in revised manuscript line 218): “Missing a sample with this mosaicism is considered to be a fundamental limitation of the current cfDNA technology because the cffDNA in maternal circulation originates mainly from cytotrophoblast (https://doi.org/10.2353/ajpath.2006.060161). In general, these samples are verified by performing a secondary test because the Z-scores are located near the boundary.”
Reviewer 2 Report
Thank you for an important manuscript. Recommend to accept.
With kindest regards,
Reviewer
Author Response
Thank you for an important manuscript. Recommend to accept.
Response: Thank you for the very positive feedback.
Round 2
Reviewer 1 Report
The authors sufficiently replied to my revisions.
I request additional comment regardins the secondary test they used:
The population of the 1,698 normal samples have been selected based on the following criteria: confirmed by an independent tool as showing no chromosomal abnormalities and confirmed as normal on the secondary test
Author Response
Thank you for the thoughtful comment on our manuscript. We have revised the manuscript accordingly.
Comments and Suggestions for Authors
The authors sufficiently replied to my revisions.
I request additional comment regardins the secondary test they used:
The population of the 1,698 normal samples have been selected based on the following criteria: confirmed by an independent tool as showing no chromosomal abnormalities and confirmed as normal on the secondary test
Response: Thanks for a thoughtful comment. As mentioned in the Result section (in original manuscript line 92), a secondary test such as CVS and amniocentesis was conducted to confirm chromosomal abnormalities. So we modified the sentences to Result section: “These normal samples were selected based on the following criteria: confirmed by an independent tool as showing no chromosomal abnormalities and confirmed as normal on secondary tests such as CVS and amniocentesis. Among the test samples, 5 and 33 samples were identified to have trisomy 18 and 21, respectively, through the secondary test.”